# Bacterial Cellulose as a Versatile Biomaterial for Wound Dressing Application

**DOI:** 10.3390/molecules27175580

**Published:** 2022-08-30

**Authors:** Julia Didier Pedrosa de Amorim, Claudio José Galdino da Silva Junior, Alexandre D’Lamare Maia de Medeiros, Helenise Almeida do Nascimento, Mirella Sarubbo, Thiago Pettrus Maia de Medeiros, Andréa Fernanda de Santana Costa, Leonie Asfora Sarubbo

**Affiliations:** 1Rede Nordeste de Biotecnologia (RENORBIO), Universidade Federal Rural de Pernambuco (UFRPE), Rua Dom Manuel de Medeiros, Dois Irmãos, Recife CEP 52171-900, Brazil; 2Instituto Avançado de Tecnologia e Inovação (IATI), Rua Potyra, n. 31, Prado, Recife CEP 50751-310, Brazil; 3Centro de Tecnologia e Geociências, Departamento de Engenharia Química, Universidade Federal de Pernambuco (UFPE), Cidade Universitária, s/n, Recife CEP 50740-540, Brazil; 4Centro Universitário Brasileiro (UNIBRA), Rua Padre Inglês, 257, R. Padre Inglês, 356-Boa Vista, Recife CEP 50050-230, Brazil; 5Centro de Comunicação e Design, Centro Acadêmico da Região Agreste, Universidade Federal de Pernambuco (UFPE), BR 104, Km 59, s/n, Nova Caruaru, Caruaru CEP 50670-90, Brazil; 6Escola Icam Tech, Universidade Católica de Pernambuco (UNICAP), Rua do Príncipe, n. 526, Boa Vista, Recife CEP 50050-900, Brazil

**Keywords:** bacterial cellulose, polymer composites, biomedical application, biotechnology

## Abstract

Chronic ulcers are among the main causes of morbidity and mortality due to the high probability of infection and sepsis and therefore exert a significant impact on public health resources. Numerous types of dressings are used for the treatment of skin ulcers-each with different advantages and disadvantages. Bacterial cellulose (BC) has received enormous interest in the cosmetic, pharmaceutical, and medical fields due to its biological, physical, and mechanical characteristics, which enable the creation of polymer composites and blends with broad applications. In the medical field, BC was at first used in wound dressings, tissue regeneration, and artificial blood vessels. This material is suitable for treating various skin diseases due its considerable fluid retention and medication loading properties. BC membranes are used as a temporary dressing for skin treatments due to their excellent fit to the body, reduction in pain, and acceleration of epithelial regeneration. BC-based composites and blends have been evaluated and synthesized both in vitro and in vivo to create an ideal microenvironment for wound healing. This review describes different methods of producing and handling BC for use in the medical field and highlights the qualities of BC in detail with emphasis on biomedical reports that demonstrate its utility. Moreover, it gives an account of biomedical applications, especially for tissue engineering and wound dressing materials reported until date. This review also includes patents of BC applied as a wound dressing material.

## 1. Introduction

Vegetable cellulose (VC) has played a remarkable role in human development and is often used as an energy source as well as for the production of materials and many other applications [1]. However, the use and production of VC raises several issues, especially those related to the pulping and bleaching cycle [2]. Industrial VC effluents are a major source of pollution and purification on the industrial scale requires the considerable consumption of electricity (approximately 1000 kWh/ton) [3]. Thus, an eco-friendly alternative is a subject of interest for industries in all fields.

Although bacterial cellulose (BC) and VC have the same chemical structure, BC offers some advantages. BC is pure cellulose, meaning that it has no linkage with pectin, lignin or hemicellulose. Thus, no energy consumption is required for further refinement. Depending on the production method, BC can be manufactured in the form of membranes, pellicles, hydrogels and granules [4], providing environmentally friendly option with numerous applications. BC possesses superb mechanical and physicochemical properties, a large surface area, a good capacity to absorb great quantities of polar liquids, high porosity, biocompatibility, non-toxicity, chemical inertness, biodegradability, capacity to form films and to stabilize emulsions [4]. Due to its suitable characteristics, this ‘green’ material can be combined with other materials and bioactive compounds to develop new biotechnological applications.

Moreover, works have been carried out on the development of nanotechnology with BC for use in cosmetics [5], effluent filtration [6,7], as well as textile [8], biomedical [9], packaging [10], and magnetic [11] applications, among others. BC has characteristics that combine fundamental nanofibrillar and macromolecular properties for applications both in vitro and in vivo in the biomedical field [12,13]. Such properties make BC nanofibrillar membranes, nanoparticles, and crystals prominent novel biomaterials.

The methodology for this paper review was the usage of the database available on Science Direct and Google Scholar search engines, with the use of ‘bacterial cellulose’, ’medicine’, ‘biomedicine’, ‘wound dressing’, ‘biomaterials’ as keywords in a combined manner. The publications time interval initiated in 2003 (year that represented a start on the subject relevance) until August 2022. Nowadays, the development of new ‘green’ materials using sustainable technologies represents a challenge for the pharmaceutical and biomedical fields. Therefore, in recent years, studies have been conducted for the production and enhancement of polymeric composites and blends based on BC for biomedical applications, with the production on novel tailored biomaterials. Innumerous reviews on using BC for medical applications have been reported [14,15,16]. However, as the number of publications has grown significantly, there is a need for new reviews synthesizing the innovative research on the subject. The present review offers a summary of advances in the use of BC in composites and polymeric blends for drug delivery systems and wound healing. BC production methods and BC-producing microorganisms are also discussed.

## 2. Wound Dressing

The skin is the ultimate stratum of the human body that has the main purpose of working as a barrier protecting the body’s internal medium from the external one. It also has the responsibility in various body functions such as adjusting body temperature with water waste control, supporting nerve endings, glands, blood vessels, and many others [17]. Skin damage leads to malfunctioning of the body’s natural activities. Ulcers (or wounds) are characterized by substantial tissue loss causing the discontinuity of the skin and adjacent tissues with changes in the anatomic and physiological structure of the affected regions. The financial value of treatments for comorbidities succeeding stage IV ulcer progress exceeds the costs for early treatments [17]. Due to the high probability of infection and sepsis, and even mortality, chronic ulcers, therefore, exert a significant impact on public health resources [18]. The most common ulcers have a surgical, venous (stasis, varicose), arterial, or neurotrophic (Hansen’s disease, diabetes, and alcoholism) origin or are the result of prolonged pressure (bedsores) [19].

With the occurrence of a wound, the cells of the human body, both individually and in groups, interact either through direct contact or the release of signals at the wound site. All cells at the injury site produce signalling molecules. Immune cells, keratinocytes, endothelial cells, and fibroblasts contribute to wound healing and assist in hemostasis and the production of vascular endothelial growth factor [20].

Wound healing can be considered as a sophisticated process, that involves several circumstances, such as coagulation, inflammation, cell division and migration, development of connective tissues and blood vessels (depending on the wound’s degree), growth of extracellular matrix (ECM), and epithelium maturity [21]. Moreover, ECMs regulates several biological functions, such as cell adhesion and their proliferation, cell differentiation, migration, their interactions, and intracellular signaling. The main cellular source of the ECM are the dermal fibroblasts (dFBs). When an injury occurs, dFBs migrate into the wound granulation tissue and transdifferentiate into myofibroblasts, the latter has the objective of dermal ECM regeneration and deposition [18].

When an injury occurs, the stem cells oversee the skin’s self-renewal and repair during homeostasis. Upon wounding, there are four phases that consists of the skin’s healing process. Those are: homeostasis, inflammation, proliferation, and remodeling, often resulting in fibrosis and scarring [22]. The principal purpose of homeostasis is the protection of the vascular system. Therefore, it prevents excessive blood loss and subsequent possible loss of organ function, and to achieve tissue integrity [23]. The inflammatory phase is responsible for establishing an immune barrier to microorganisms’ contamination and to eventually destroy those that have been introduced into the wound during injury [24]. The proliferative phase focuses on the re-establishment of the epithelial barrier by wound contraction through different processes, such as angiogenesis (or neovascularization, to the newly generated tissues and ECM), fibroplasia, and epithelialization [24], following the re-establishment of functional microvasculature (arterioles, capillaries, and venules) and the elimination harmful microorganism. Dermal and epidermal cellular regeneration can occur just before the wound’s closure and possible scar formation and depending on the injury’s degree, this proceeding (named maturation or remodeling) can take a number of months [25].

When a wound occurs, other events also take place. Coagulation involves platelet aggregation, providing a temporary ECM for cell migration [26]. The inflammation process takes place where white corpuscles cells such as phagocytic cells (neutrophils and monocytes) and macrophages migrate to the injured wound area [27]. During this period, damaged cells, pathogens, foreign particles, and microorganisms are removed from the wound site. The tissue remodeling phase involves the strengthening of the tissue through the synthesis of collagen and elastin. During this step, the wound keeps on contracting, and fibers are reorganized, forming a scar [21].

Numerous types of dressings are used for the treatment of skin ulcers-each with different advantages and disadvantages. The choice of dressing is based on factors such as effectiveness, safety, and cost in achieving a satisfactory result, as most ulcers have a slow healing process [28].

According to Santos et al. [29] the most widely used dressings for the treatment of these skin conditions are traditional gauzes or gauzes with petrolatum and/or paraffin, dressings with calcium alginate or silver nanoparticles, polyurethane films and hydrogels. However, many of these dressings are expensive and, in some cases, can exert a detrimental impact on the healing operation due to infection or the occurrence of secondary wounds. Thus, the choice of dressing is dependent on the wound’s classification and each specific phase of the healing process.

The human skin offers protection from the invasion of exogenous substances and harmful microorganisms, such as fungi, bacteria, and viruses. It is also responsible for stabilizing the internal environment in the body [30]. The objective of wound healing enhancement is to quicken the wound repair, thereby preventing infections, reducing pain, removing dead tissue, supplying a humid environment, diminishing edemas, and increasing the flow of blood. Due to the physical, social, and financial impact of chronic ulcers and the difficulty in curing these skin problems with conventional treatments, researchers have recently been focusing on the progress of novel therapeutic procedures for wound treatments [31]. Despite several studies being conducted in the recent past working toward the development of a ‘perfect’ wound dressing biomaterial, there are no currently available materials that entirely accomplish the needed characteristics for a quick and enhanced injured tissue recovery. With that being said, the search for wound-dressing skin substitutes based on biopolymers remains a challenge as an interesting substitute [32].

## 3. Biocompatibility

The definition of biocompatibility is the capacity of a material to execute its desired functions with regards to therapy and induce appropriate responses from the patient in a specific application. It is also the interaction with living systems without causing harm, toxicity or any undesirable local or systemic effects [33]. In 2010, Kohane and Langer [34] defined biocompatibility as “the benignity of the relationship between a material and its biological environment”.

Biomaterial is another widely used term. By definition, biomaterials are designed to play a directing role in therapeutic procedures or diagnostic methods through interactions with living systems [33]. In the field of medicine, biomaterials must be biocompatible, meaning that such materials cannot cause inflammation or any adverse tissue reaction [33,35]. High biocompatibility is attained when a material interacts with the body without inducing unwanted responses, e.g., immunogenic, toxic, thrombogenic, or carcinogenic [35].

However, numerous factors exert an influence on the interactions of a material with a biological environment. Diverse physical, chemical, biological, biochemical, and physiological mechanisms as well as the shape and size of the device directly influence the outcome [36]. Technological advances applied in the field of medicine must ensure the appropriate interaction of materials, particles, drugs, and active compounds in biological domains for the conduction of illnesses, as cellular and extracellular interactions can provoke a set of biological consequences. Such effects are conditional on the physicochemical characteristics of the materials employed. These determine the achievement of the intended results according to their biocompatibility. Understanding the mechanisms behind these different results enables predicting interactions between the material and biological environment [37].

Thus, biocompatibility is an essential aspect of biomedical materials, as such materials must persist in contact with living tissues in the absence of allergenic and toxic effects. To ensure the safe, effective use of materials for medical applications, a study regarding the interaction between the biological system of interest and the desired material must be performed. Moreover, research on the biocompatibility of a material should be conducted aiming at the specific environment in which the material is going to be used [37]. Such care is fundamental to minimizing the occurrence of cytotoxicity, genotoxicity, and other problems.

An important fact is that a particular biomaterial application may have an adverse repercussion on a specific type of tissue but that may not necessarily cause the same response in a different application or on a different type of tissue. Moreover, the intrinsic characteristics of biomaterials do not exclusively determine whether a certain material is biocompatible or not; they are biocompatible as a function of their specific applications. For instance, poly (lactic-co-glycolic acid) (PLGA) is rapidly eliminated from the human body and does not cause peritoneal adhesion. However, PLGA microparticles remain a longer period of time in the peritoneal cavity and can cause peritoneal adhesions [38]. Thus, there is a need to evaluate the material’s biocompatibility, especially biomaterials, on an individual basis and specifically in terms of the tissue of interest and intended application. Taking all these facts into consideration, one may assume that biocompatibility of a material is dependent on its chemical structure and formulation and is also related to the local or systemic effect on the body [38].

### 3.1. Bacterial Cellulose Biocompatibility

Cellulose-based materials can be considered biocompatible. Those cellulosic materials, such as oxidized cellulose and regenerated cellulose hydrogels, meet such requirements, and are widely used in the medical field. Such use is related to the considerable potential for biocompatibility, as the absence of genotoxicity is one characteristic of BC [39].

A work has evaluated the tissue reaction after subcutaneous implantation in mice of a BC membrane and focused on the in vivo biocompatibility of BC. The implants’ evaluation were done regarding the aspects of foreign body responses, cell ingrowth, chronic inflammation, and angiogenesis. The authors found no evidence of foreign body reaction (FBR) during the study, and the formation of new blood vessels around and inside the implanted cellulose was also observed [19]. In vivo tests were also carried out for tubular shaped BC membranes aiming for the substitution of vascular blood vessels. The results have shown similar data to the aforementioned work [40].

However, as the compatibility of biomaterials must be exerted in a controlled manner, newer studies have been focusing on the improvement of such property with the usage of cellulose composites, specially to enhance cell adhesion, as native cellulose does not promote it by itself. The biomaterial surfaces should stimulate the absorption of specific proteins and therefore promote the subsequent cellular interaction [38]. It has been demonstrated that the enhancement of cell adhesion can be achieved by immobilizing the adhesion proteins onto the biomaterial surface [41].

### 3.2. Bacterial Cellulose Composites Biocompatibility

Authors used a BC/gelatinized lotus root starch (LRS) composite to produce membranes with thicker and denser cellulose fibrils. Live/dead chondrocytes assay on BC/LRS composite revealed great cell viability of 85–90%. Through the overall results, the authors were able to show a composite with higher mechanical strength and better cell biocompatibility in contrast with BC alone. This work suggests the potential use of BC in cartilage tissue engineering [42].

Another study aimed to fabricate three-dimensional (3D) scaffolds of BC and chitosan (BC-Chi) through a one-step ex situ process. An interaction analysis was performed with ovarian cancer cell lines, and the data showed that the cells were adhered to the surface, and they infiltrated intensely into the BC-Chi matrix, demonstrating a strong cell-scaffold interaction. The fabricated scaffold with its improved biocompatibility showed its prospective applications of cancer diagnosis [43].

The development of BC/hydroxyapatite (BC/HA) nanocomposite hydrogel has also been reported as a potential 3D cell-culture platform [44]. Within their results, the arrangement of composites closely resembled the native ECM, indicating the potential to act as a substrate for cell culture. The obtained BC/HA composites exhibited enhanced moisture absorption due to their high swelling ratio. The in vitro biocompatibility results displayed a great percentage of cell proliferation. These findings indicate innovative composites with the potential of being used in 3D cell-culture applications [44].

A protein that has been used to improve BC’s biocompatibility is collagen (C). A study has prepared BC-C composites by ex situ process. The incubated fibroblast cells had a great performance regarding cell adhesion and proliferation. In contrast to pure native BC, the developed composite showed a better cytocompatibility [45]. In summary, fabricated BC composite scaffolds can represent an approach to extend the cell adhesion, growth, and transplantation of scaffold-seeded cells, thereby improving their biocompatibility. This can provide a new perspective on future research towards biomedical applications of cellulosic based materials in several medical fields.

## 4. Biopolymers

Biomaterials are substances of a natural or synthetic origin that are tolerated either temporarily or permanently by the different tissues that constitute the organs of living beings. These materials can be entirely or partially used as a system to treat, restore, or replace tissues and organs or can be used in a medical device with the intention of interacting with biological systems [46]. The study of biomaterials involves understanding the properties, characteristics, functions, and structures of biological tissues, synthetic materials, and the interactions between both.

Polymers are macromolecules made up of repeated monomeric structural units. Those of a natural origin (biopolymers) are fundamentally produced by living beings or can be obtained using raw materials from renewable sources, which are denominated “green polymers” [47]. In recent years, biopolymers have been noticed because of the possibility of developing high-performance and low environmental impact biocomposites and blends that offer the advantages of abundant availability, renewability, and eco-friendliness. The research investment of such materials has increased substantially, biopolymer composites and blends are anticipated to replace numerous conventional materials in biological, engineering, and medical applications [48]. The desired properties of biocomposites can be accomplished by blending an appropriate biopolymer with suitable additives. A set of variables related to biopolymer composites, such as chemical composition, degradation kinetics and mechanical properties, can be adapted to application needs.

High biodegradability is one of the greatest advantages of biopolymers. Decomposition in the environment occurs though the action of microorganisms in the ecosystem in which the material is discarded. Such biological activity breaks downs molecules into smaller pieces in less time, leading to a lower environmental impact [49,50]. The natural anaerobic biodegradation of diverse polymers leads to the production of methane (CH_4_), which is a harmful gas to the environment and is only slowly reabsorbed through natural environmental processes. Despite this disadvantage, biopolymers offer significant benefits [51].

Several components, such as the choice of employed raw material, method of extraction and different approaches of functionalization, are very important in the determination of the characteristics of biopolymers [52]. Biopolymer fibrils and nanofibrils have been investigated for the reinforcement of polymeric composites due to their great availability, low cost, and good mechanical properties. Such materials can be applied in a variety of emerging technological fields, like optoelectronics, energy, environmental science, and biomedicine.

### Biopolymers in Medicine

Polysaccharides can give origin to versatile materials with enhanced properties because of their delicate nature and inherit properties like bioactivity, bio-adhesion, and homogeneity [53]. Cellulose, chitosan, alginate, starch, pullulan, hyaluronic acid (HA), pectin, xanthan gum, and dextran have all been reported in medicine applications [53]. Many studies reveal that the use of polymeric blends and/or composites of the previously cited biopolymers are able to achieve adequate and enhanced characteristics for biomedical applications [54].

It is important to highlight that the use of some of these polysaccharides must be done with caution, as some of them might me allergenic to humans [55]. Therefore, the polymer combination must be performed carefully in order to minimize at most this occurrence. With their use in medicine, the chemical and physical properties are key for the promotion of cell attachment, migration, increase of cell number, and specialization of functional cells. A suitable alternative to overcome these limitations is a must [56].

Chitosan is a linear and hydrophilic polysaccharide that is commonly used for biomedical engineering applications [57]. A polycationic polysaccharide that is also used is alginate. However, this biopolymer has the disadvantage of a difficult degradation [58]. Starch, pullulan, and xanthan gum are hydrophilic substances, with high polymeric branches that are also widely used. HA is a non-sulfated glycosaminoglycan naturally dispersed throughout the ECM of loose, dense, and specialized connective tissues. Nonetheless, it is a polymer occurring in a vast number of configurations and shapes and depending on the application, it has the disadvantage of a low half-life [59]. However, all specified polymers of natural sources have limitations due to their weak mechanical properties.

The biopolymer’s intrinsic biological and physicochemical properties and structural characteristics, being large surface area, high porosity, and low density make these materials suitable for applications as biomaterials used in regenerative medicine [60], tissue engineering [61], drug delivery [62], wound dressings [63], anti-cancer treatments [64], antimicrobial agents [5], the reduction of obesity [65], biosensors [66], etc. Such versatility is due to the ease of processing biopolymers in different formats, such as sponges, scaffolds, hydrogels, and membranes [67]. Some factors related to biopolymers integration and their exploitation with the human body still exhibits constant improvements [53]. Consequently, the necessity to acquire novel materials represents a continuous necessity.

Following continuous research, diverse proceedings have been developed for their application in biomedicine. As cellulose has a non-allergenic profile, low immunogenicity, and excellent cytocompatibility, BC is considered one of the biopolymers of interest for the fabrication of biomaterials. BC has been widely explored due to its combination of properties and characteristics, which are described in greater detail in the following section. Such features enable the use of BC in different fields, particularly in biomedical applications [68].

## 5. Bacterial Cellulose

Cellulose occupies the first position among the most abundant biomasses found in nature [69] and has two native forms: (1) pure cellulose, which is obtained directly from its natural state, such as BC and that produced by some algae species, and (2) complex cellulose, which contains impurities, such as lignin, pectin, and hemicellulose [70].

First reported by A. G. Brown in 1886 [71], BC consists of a translucent, gelatinous film composed of micro and nano fibrillary cellulose distributed in unsystematical directions (Figure 1).

This biopolymer is produced extracellularly, through an aerobic process, which can be of cell-free enzyme systems [72,73], by acetic-acid Gram-negative bacterial cultures of *Aerobacter*, *Agrobacterium*, *Komagataeibacter*, *Pseudomonas*, *Achromobacter*, *Azobacter*, *Rhizobium*, *Salmonella*, and *Alcaligenes* and Gram-positive bacterium *Sarcina ventriculi*. These aerobic and non-photosynthetic bacteria, or aerobes are generally found in alcoholic beverages, vinegar, fruits, and vegetables [74]. The most efficient BC producer belongs to the genus *Komagataeibacter* (previously called *Gluconacetobacter*) (Figure 1b.) due to its greater production capacity and ability to grow in media with a diversity of carbon/nitrogen sources [75].

The use of a system free of cells is a possible cellulose production strategy, as it can improve the fiber strength and density. This process provides the carbon source uniquely for cellulosic production and for a long period, when compared with the production by microbial cell system [72,73]. The literature shows that the BC production in a cell-free system can operate even in anaerobic conditions. Ullah et al. [72,73] concluded that higher carbon source availability content (such as dextrose) in the production medium for longer fermentation time favours the microfibrillar synthesis by the cell-free system, in contrast to a bacterial cell system. This prolonged the synthesis of cellulose, resulting in larger pore diameter and a more compact cellulosic film fibrillar arrangement. The cell-free cellulose showed lower values for its crystallinity degree, a lower water release rate, higher tensile strength, a slightly lower elongation at break (strain), and higher thermal stability [72,73]. Such characteristics must be taken into consideration according to the intended application.

This unique nanofibrilated matrix (Figure 1c) is being extensively investigated for various applications. As it can be produced in several configurations, it is highly versatile polymer. The intrinsic characteristics of BC, namely its biodegradability, mechanical strength, biocompatibility, haemocompatibility, micro and nanoporosity, and its distinctive surface chemistry, show how this biopolymer satisfy the demands for a great many biomedical applications.

Inter and intramolecular covalent hydrogen bonds in the hydroxyl (–OH) group of the cellulose chain impede the solubility of cellulose in water. Such bonds play a substantial function as a ligand that maintains the polymeric chains of cellulose together, thereby conferring to the cellulose matrix’s a high tensile strength [67]. This makes a material adequate for tissue engineering applications, as the material is able to maintain its structure even if applied to the human body’s natural physiological conditions. BC’s great capacity of water absorption and strong mechanical properties are excellent attributes for biomedical applications [67].

### 5.1. Bacterial Cellulose Synthesis and Production

BC is predominantly produced from C sources, mainly being glucose. However, other carbohydrate sources have also been reported, such as fructose [76], sucrose [77], galactose [78], corn steep liquor [77], and agro-industrial by products [79,80]. This biopolymer is produced via a progression of microbial enzymatic reactions, where the conversion of sugars into dextrose happens and then occurs their polymerization into cellulosic chains [81]. Molecular studies revealed the collaboration in the supramolecular assembly of cellulose fibrils of certain genetic operons in the cellulosic biosynthesis and extracellular transport. The structural and functional properties of BC, and its production yield have the possibility of genetic and metabolic modelling, through genome sequencing of the various BC-producing strains, this allows a functionalization for multipurpose applications [82].

The biosynthesis of BC serves various purposes for the microorganism, such as the protection of physiological, mechanical, and chemical stability, the enhancement of interactions and nutrient diffusion [83]. BC is produced by a biochemical process through oxidative fermentation in both non-synthetic and synthetic media, with the control of specific enzymes. Production starts in the microbial cells, cytoplasm, with the synthesis of β-1,4-glucan chains. These chains crystalize and form microfibrils that will later produce small pellicles that will then form membranes [70]. In a more detailed explanation, the biosynthesis pathway for BC initiates with the isomerization of a glucose sugar phosphorylated at the hydroxy group on carbon 6 molecule into the same molecule with a -OH group on carbon 1. Afterwards, it reacts with uridine-5′-triphosphate, forming uridine-50-diphosphate-alfa-D-glucose, that is later polymerized into 1,4 glucan chains in a linear conformation. The recently produced cellulose polymer chains are then secreted across the bacterial cell wall [84,85].

BC’s production is usually achieved in a mainly C and N nutrient-rich fermentation media either in agitated or static manner, or in a bioreactor, as shown in Table 1. These methods give rise to structures with different properties and morphologies. The choice of production method depends on the intended application.

Production in a static culture result in a BC membrane, whereas an agitated culture results in the formation of suspended fibers in the form of irregular pellets [90]. Static BC production takes place at the air-liquid medium interface, whilst by agitated production, the pellets are formed submerged in the liquid fermentation medium (Figure 2). The usage or more complex techniques can result in cellulose with different morphologies, such as hollow spheres [91], aerogels [92] and even a water-in-oil emulsion [93]. BC’s resulting properties, micro and nanostructure, and morphology of BC are different. The production method of choice is conditional on the ultimate BC’s applications. However, the static culture method for production of BC still widely remains the selected approach.

In order to reduce BC production costs, there have been works focusing on the possibility and practicability of industrial, mainly agricultural-based residues and wastes being used as nutrient sources [76]. The BC from alternative mediums have similar physicochemical properties as those produced from the standard HS medium.

The unique synthesis process gives BC highly desirable physicochemical, structural, and biological properties, such as high purity, a high degree of polymerization, high hydrophilicity, high crystallinity (ranging from 60% to 90%) and even ex situ modifications into alternative formats [94,95,96]. Studies on the growth kinetics of microorganisms capable of producing BC membranes assist in the obtainment of specific thicknesses through changes to the production media or growth conditions, thereby facilitating the use of BC for the most diverse applications [97,98].

### 5.2. Bacterial Cellulose Applications

Unlike other types of synthetic membranes and even other types of biopolymers, the unique characteristics, properties and versatility make BC a biopolymer with considerable potential to be exploited by biotechnology in broad range of applications, such as medicine [67], foods [99], surgical and protection masks [100], cosmetics [5], electronics [101,102] and packaging [10,103], as well as for the separation of oily mixtures [7,104], among other fields.

BC can also be adapted to other applications through chemical modification or the functionalization of synthesized nanocellulose [104,105]. Such methods involve the modification of BC’s functional groups located on the surface of the membrane or pellets through different approaches, such as the addition of ionic charges by amidation, phosphorylation, acetylation, sulfonation, oxidation, carboxymethylation, etherification, silylation, etc. [106]. The benefits of the use of BC and its products for the treatment of wounds and infections and for patient recovery can revolutionize the biomedical field.

### 5.3. Bacterial Cellulose in Biomedicine

BC is a nontoxic, biocompatible, moldable, highly absorbent biopolymer [107]. Despite its promising chemical and physical properties, biomedical applications are limited because BC naturally lacks antimicrobial activity. However, studies have been conducted to overcome this problem with the incorporation of active molecules and materials with antimicrobial properties into the cellulosic matrix via several in situ and/or ex situ strategies for the development of novel polymeric materials [2,17,108,109,110].

BC’s fibrillar structure with its nanoscale and high porosity is an appropriate macromolecular support for the absorption and adsorption of active substances, pharmaceuticals, and drugs. As it has a neutral electrostatic charge, this facilitates the incorporation of bioactive compounds with both negative and positive charges [15], facilitating their incorporation onto the polymeric matrix. Therefore, they are ideal for the development of innovative specific controlled release systems, especially in biomedical engineering, including wound dressing and transdermal drug delivery systems [107]. In addition, when used as a membrane, it can contribute to an increase in cellular adhesion as well as the proliferation, migration, and differentiation of cells, thereby accelerating the re-epithelialization, which results in a faster wound healing process [15,111]. As previously stated, its performance as a biomaterial has attracted attention for use in drug delivery systems [15,111,112,113], wound dressings [114], tissue scaffolds [115], and implants [116].

Due to its unique characteristics previously mentioned, BC used in skin wound treatment has great potential. Studies offer promising results regarding the modification of BC to give it antimicrobial properties. The different modifications include the incorporation of antibiotics (such as Bacitracin and Amoxicillin) [15], silver nanoparticles [117,118], copper nanoparticles [119,120], silver chloride nanoparticles [121], silver montmorillonite nanoparticles [122], impregnation of montmorillonite nanocomposites [123], benzalkonium chloride solution [124] the use of immobilized lysozyme in bacterial cellulose nanofibers [125], the incorporation of propolis extract [5] and the creation of a BC–chitosan composite [123]. BC can also be applied as scaffolds for seeding of cells. The literature has shown several types of cells that can grow in the presence of BC [47,126,127,128,129].

One of the proliferation phases that occur following a wound consists mainly of the migration of fibroblasts from different sources, which results in the development of new connective tissue and microscopic blood vessels (granulation tissue). Fibroblasts are responsible for repairing damaged tissue by providing a new extracellular matrix, which is followed by the closure of the wound [130]. Studies have investigated the combination of cellulose and other biomaterials to enhance the proliferation, infiltration, and adhesion of fibroblasts [46,120,131].

As mentioned above, BC can be used for the development of innovative materials. Specially designed materials whether of a synthetic or natural origin that regulate the environmental conditions of a wound are at the forefront of regenerative medicine [111]. 

The fabrication of BC-based biomaterials includes biological, chemical, or physical methods to enhance the properties of these materials for application in a specific field [132]. A considerable number of studies have been carried out to improve the BC properties to enable its use in biomedicine by incorporating other polymers [30], nanoparticles [109], active molecules, or extracts [5].

To gain a better understanding of the biocompatibility potential of BC, Pértile et al. [39] surgically performed the subcutaneous implantation of BC in rats to provide greater contact with biological tissues in comparison to wound dressings. The results demonstrated that the BC structure exerted a positive influence on cell invasion and the behavior of the implant over time. The macroscopic examination revealed that the BC implants maintained their shape, but internal fissures lined with migratory stem cells were evident in the histological analysis. The authors found an absence of clinical signs of inflammation at the incision sites. The cellular response evolved progressively to chronicity, with a reduction in inflammatory cells around the implants and a predominance of macrophages over neutrophils. After three months, macrophages, fibroblasts and endothelial cells were predominantly found on the implants. All animals implanted with cellulose nanofibers survived and presented development throughout the period studied. As expected, the BC implants in this experiment did not provoke an uncommon response of body systems. Thus, it was clear that BC did not cause a foreign-body reaction beyond the formation of a thin fibrous layer [39].

BC has also been used in biomedical tissue engineering and bone grafting. Such applications are possible due to the composition of bones, which consists of a inorganic mineral phase (also called (also called inorganic bone phase, bone salt, or bone apatite, which consists mainly of calcium hydroxyapatite) and an organic phase (composed mostly of collagen and non-collagen proteins) [133]. Thus, BC can be used as a matrix for the obtainment of calcium carbonate crystals, which are believed to improve biocompatibility. Studies conducted by Stoica-Guzun et al. [133] revealed that BC nanofibrils can reproduce the characteristics of collagen nanofibrils for the deposition of calcium and phosphorus through biomineralization.

Studies have also evaluated the use of BC for cardiovascular applications, with BC used in the production of blood vessels in in vivo tests. These “synthetic veins” were developed by Klemm et al. [1] and used to repair the carotid artery in a rat. After one month, the BC/carotid artery complex was found to be enfolded with connective tissue, demonstrating BC’s potential as replacement of blood vessels. BC have also been reported as an advanced biosensing [134,135] and diagnostic materials [43].

Considering the properties of BC and possible modifications and functionalization to obtain novel biomedical functions, this biopolymer has considerable potential in the treatment of wounds. Indeed, BC demonstrates superiority over conventional dressings due to its easy attachment to the skin without restricting the movements of the patient. Thus, microbial cellulose has been increasingly used for wound dressings and tissue engineering applications [136,137].

#### 5.3.1. Functionalized Bacterial Cellulose for Biomedical Applications

According to Lima-Júnior et al. [138], wound dressings need to be effective, offer practical applicability and be easy to produce, obtain and market. Thus, the materials that compose dressings need to be inexpensive, enable easy storage and prolonged stability, must not have antigenicity, should have good flexibility and stretch resistance, offer good adherence to the wound site, as well as good adaptation around the wound while facilitating joint movements. Such materials should also be applied in a single surgical session, offer easy handling, attenuate pain, accompany body growth and maintain body temperature. BC is compatible with such characteristics, which suggests the considerable potential of this biopolymer for applicability as a biomaterial in medical applications [138]. The characteristics that are essential for optimal wound dressing is thermal insulation, biocompatibility, cost efficacy, mechanical stability, non-toxicity, to maintain a moist wound environment, infection prevention, and adequate gaseous exchange [138]. All the aforementioned characteristics are inherited to BC.

During the wound treatment process, proper moisture is required to enable a shorter recovery time. The high water-holding capacity enables microbial cellulose to conserve the ideal moisture of the skin’s wound and even ulcers site. Due to the nanofibril network, these membranes can have additives incorporated into the matrix and create a physical protection that impedes the infiltration of microorganisms, thereby diminishing the risk of infection while also reducing both pain and healing time [139] as shown in Figure 3.

As specified by Czaja et al. [140], BC’s biocompatibility for bandages and dressings is related to its distinct structure, which serves as an adequate environment for wound healing. This nanofibrillar structure is able to aid in eliminating the discomfort symptoms by increasing the adsorption of exudate from the wound and isolating the skin’s nerve endings. Excessive exudate results in the separation of tissue layers, which hinders the healing and tissue regeneration process. Thus, the adequate absorption of exudate is an important aspect in the development of modern dressings [140]. In comparison to conventional dressings, such as moist and ointment-impregnated gauze, BC enables a faster skin healing process. Moreover, BC has shown good cytological and histological compatibilities [141], diminishing the risk of infection and sepsis.

A balance is needed between the absorption and adsorption of these fluids and the release of moisture, as the dehydration of the wound surface hinders the successful recovery of the tissues. Because of its high water-holding capacity (WHC) and water/moisture release rate (WRR), BC is a material with great potential for wound dressing applications. Moreover, its membranes structures can be adapted to diverse situations in this type of application [142]. Studies have demonstrated that BC-based linings diminish the pain of the wound, quickens the re-epithelialization, and lessens would infection rates and scars [19,69,118,131].

The main characteristics of currently available wound treatment materials are good absorption and permeability. However, these materials often cause trauma and harm upon removal from the wound site. Comparing the properties of BC to conventional materials used in the treatment of wounds, membranes produced by microbiological fermentation can be used directly after the fermentative process following rinsing with running water. BC membranes can also be processed in different forms suitable for various wound dressing applications, as previously mentioned [136].

Another important characteristic of wound dressings is the capacity to remain structurally intact between placement and removal, especially when placed close to a joint, as the movement of the body can lead to the exposure of the wound. The tensile strength of BC membranes is an important factor and depends on both the culture conditions and treatment. Tensile strength can reach 260 MPa, with stretching up to 32% prior to breaking. Such mechanical properties of strength and flexibility demonstrate that BC is adequate for a variety of dressings in different types of treatment and sites [136,142,143]. 

Depending on the usage of the BC’s polymeric matrix, it is interesting to modify its porosity, as appropriate porosity such characteristic of the biomaterial needs to be similar to the replaced tissue that is going to take place. Nicoara et al. [111], modified BC through in situ and ex situ and obtained a BC/Hydroxyapatite (HAp) composite with the incorporation of silver nanoparticles (AgNPs) with 10–70 nm size. All obtained materials demonstrated a homogenous porous structure and high-water absorption capacity, <5% degradation rates in artificial human blood plasma, and good antimicrobial action due to the AgNPs. The obtained composite’s prepared via in situ showed a wider porosity distribution and better homogeneity [107]. Other innovative approaches to BC’s modification have been widely reported in literature [139], including 3D-bioprinting. Studies show bioprintable BC as a medical material to be utilized in different tissues and scaffolds [44,61,98,114,128].

According to Khan et al. [68], when used in the treatment of wounds, the characteristics of BC can be classified according to the additional, intrinsic, and improved properties of the membranes, demonstrated in Figure 4.

Traditional bandages, such as gauze dressings, are convenient for drug delivery. However, such dressings adhere easily to the exudate, causing secondary wounds and even infection [144]. Infection prevention should be of low cost to assist in wound healing and to be easily removed [114]. Modern dressings have been produced with the objective of reducing inflammatory and immunological diseases while also preventing dehydration and enhancing the healing process [132]. Several hydrogel polymers that can be easily manufactured have been used for wound dressings, such as collagen [108], alginate [145], cellulose [146], and composites [112].

BC is an inherit occurring material with nanoporosity, and this highly desired property has attracted scientists as shown by the increasing numbers of annual publishing that appear in ‘Google Scholar’ involving the descriptors ‘bacterial cellulose’ and ‘medicine’ and ‘bacterial cellulose’ and ‘wound dressing’ or their combinations (Figure 5) as the search words.

Due to BC’s polar molecules and porous geometry, the cellulosic matrix has been widely processed with other materials and polymers resulting in blends and composites for targeted applications, as demonstrated in the patent works in Table 2. The coalescence of additives onto its polymeric chains has led to unique features, such as transparency [147] bactericidal activity [10], and enhanced biocompatibility [148,149].

Literature shows other very interesting manuscripts on BC modifications. Some of which are: In situ and ex situ obtention of BC/Hydroxyapatite (HAp) composite incorporated with AgNPs [111]; Printable BC/polycaprolactone (PCL) composite loaded with antibiotics [113]; In situ composite of transparent antimicrobial AgNPs/BC films [150]; BC whiskers and poly (2-hydroxyethyl methacrylate) (pHEMA) hydrogel incorporated with AgNPs via ex situ [151]; BC/gelatin (Gel) membrane guided with electrofield (EF) stimulation [152]; BC/Gel/selenium nanoparticles (SeNPs) in situ nanocomposite hydrogel synthesis [153]; BC impregnated via ex situ with antibacterial bioactive extracts [154]; Addition of semi-dissolving microneedles and TEMPO-oxidized BC nanofibers [155]; Nanopolymer blend of BC and polyacrylamide mesh [156]; BC polymeric blend with low molecular weight deacetylated chitin biopolymer [157]; BC membrane reticulated with citric acid and additivated with inorganic catalysts [158], and curcumin-loaded BC nanocomposite prepared by ex situ method [159].
molecules-27-05580-t002_Table 2Table 2Studies and patents related to modifications of bacterial cellulose for use as wound dressing.TitleBC ModificationApplicationPatent CN103861146A [160]In situ polymer modifications to confer sticking properties to the BC, particulate leaching was carried out to make the film more porous.Manufacture of a BC patch with great biological compatibility, excellent mechanical, anti-adherent and antimicrobial properties in environments of moist soft tissues. Patent CN104403136 [161]BC and pectin composite formed through in situ process.The pectin/BC composite film presented 29% greater transparency, less porosity and permeability to water vapor as well as excellent sealing, enabling a safe, non-toxic wound dressing with considerable potential for applications in the field of medicine.Patent CN106074458 [162]BC processing with polyacrylonitrile, resulting in a polymeric composite.Preparation of a fibrous polymeric composite with the capacity to deliver anti-inflammatory drugs for transdermal administration. This composite-drug combination can be used to achieve an appropriate, controlled drug-release rate.Patent CN109966566A [163]Dual-layering of BC in nanoporous and modified (in situ) microporous structure.Preparation of a BC transdermal patch that can be used for wound repair. The inventors state that the patch has the potential to provide the basis for studies on hernia repair, cartilage scaffolding and other biological materials.


BC’s porous network exerts a positive response human body’s cell. That is, dressing materials with adequate moisture degree are able to accelerate the wound healing procedure and to protect from eventual microbial contamination. Thus, the ability to manage BC’s porosity can be utilized to modify its WHC and WRR. Both parameters are essential in determining the applicability of BC as wound dressing material [162,163,164].

According to Dahman (2009) [165], the hydrophilic polarity, high number of free fibrils and high surface area are responsible for BC’s high WHC, reaching 100–200 times its dry weight. This characteristic makes microbial cellulose a successful material for burn and scalds treatments, assisting the skin’s thermoregulation of surface moisture content [166].

Another important point is the purification of BC-based dressings. Efficient purification of BC’s raw material must be performed to guarantee the complete removal of residues from the culture medium and bacterial cells that can cause contamination during the use of the product. Generally, the thermal purification of the membrane with a basic pH solution, such as NaOH solution between 0.1 and 1M at 60–100 °C for 1–3 h, followed by pH neutralization with the aid of organic acids, or rinsing in running water until achieving the desired pH [140].

To ensure biosafety in this and other fields of application, Nascimento et al. [167], conducted a study to determine whether gamma irradiation could be used for a simple, effective sterilization of the BC membranes. Gamma irradiation is often used as a sterilization method for medical products and equipment. However, due to its high penetration power, it was necessary to assess its reactions on the physicochemical and structural properties of the membranes. The researchers used cobalt-60 as the irradiation source. The results demonstrated that gamma irradiation (at 25 kGy) did not cause any relevant alterations to the polymeric properties of membranes and therefore constitutes an effective sterilization method for this material.

Active principles that inhibit microbial growth represent another crucial characteristic for dressing used in the treatment of wounds, chronic ulcers, and burns. The antimicrobial function can be added through a modification of the structure and impregnation with antimicrobial agents, such as biopolymers [168], cationic antiseptics [169], antimicrobial peptides [170], antibiotics [171], natural active compounds [5], or inorganic nanocomposites with bactericidal properties [172].

Some works that study BC for transdermal drug delivery include cellulose membranes embedded with anti-inflammatory drugs [173], reducing agent compounds [174], nanoparticles [175], etc., aiming for similar human skin permeation rates to the already commercialized patches.

#### 5.3.2. Bacterial Cellulose in Medicine

According to the American Academy of Alternative Routes of Drug Administration [176], drugs can be administered to the human body through different anatomic routes. The chosen used material conditions the most adequate route of administration, and that is essential to ensuring the success of the therapeutic process.

Studies on polymers of a natural and artificial origin have demonstrated different drug delivery systems (DDS) ensuring minimal or no side effects. Nanotechnology used jointly with nanomembranes is a novel, promising strategy for DDS [177]. According to the authors cited, the nanocarriers’ properties responsible for enhancing the efficiency of DDS include biocompatibility, biosafety, encapsulation capacity, molecular polarity, bioavailability, and therapeutic efficiency (controlled distribution and release, cellular absorption, excretion, pharmacokinetics, toxicity and depuration).

The development of biomaterials that enable the controlled release of medications is of considerable interest, as the administration of medications in pure form has undesirable effects, such as the rapid degradation of substances in the organism, distribution to non-target tissues and organs and a possible significant reduction in the effective concentration in the target tissue. Moreover, the combination of these factors can lead to systemic toxicity [176]. The controlled administration of medications enables the extension of treatment with the use of controlled concentrations that reduce the risk of irritation and enable the use of substances with a short biological half-life [176].

Different materials can be used as the base, vehicle or matrix for the development of controlled DDS. BC is a viable option for this purpose and has been used as a topical agent for the encapsulation and delivery of different types of active compounds, including insoluble drugs [176,178]. Recent advances in the use of BC for controlled DDS include oral, ocular, intratumor, topical and transdermal delivery. Besides its use as a transporting system, BC can also be employed to encapsulate drug excipients, such as thickeners, emulsifiers, stabilizers and surfactants [179].

The drug release process is controlled by diffusion, which depends directly on pH and can result in different responses that can be adapted through the use of physical treatment or chemical modification [180]. BC can modify the drug release process through water retention, enhanced adhesion, or the formulation of a film [179].

#### 5.3.3. Bacterial Cellulose for Drug Delivery

Hydrogels are materials with 3D network formed by cross-linked hydrophilic polymers. These materials have great uptake/holding and release capabilities for water and other polar fluids [180,181,182,183,184]. Amidst the several polymeric materials, nanocellulose-based hydrogels have received attention [184].

At the end of the 20th Century, BC was used for the first time as non-permanent skin replacement and under commercialization named BioFill^®^, currently known as Dermafill™. The product is made of partially dried BC membranes for the treatment of damaged skin by thermal burns, abrasions, lacerations and ulcers. The performance of the dressing was better than conventional dressing with regards to pain relief and the acceleration of the healing process [35]. Several other BC byproducts have since become commercially available for topical application for wound healing [136], such as Bionext^®^, Bioprocess^®^, and XCell^®^, as demonstrated in Table 3.

These dressings imitate the extracellular matrix, enhancing epithelialization and the regeneration of tissues better than traditionally uses synthetic products and conventional gauze. The performance of these materials has been confirmed in different types of treatments, such as chronic ulcers and burns [15].

## 6. Future Perspectives and Challenges

BC is a biomaterial of high versatility for different applications, including foods, cosmetics and products of personal care, the biomedical industry, the textile industry, and other biotechnological sectors. Such diversification in the applicability of BC is due to its unique properties, production ease, and the simplicity of its functionalization processes. However, several limitations remain that restrict the more diversified application of BC.

A considerable challenge to overcome with the use of this material is the discovery of a cheap, adequate carbon source that does not compromise the production of food products. Greater effort is needed in the early stages of its production. Moreover, the productivity and specific characteristics of BC various microorganisms’ species and strains, in addition to different carbon sources used, underscoring the need for innovative research in this field.

The greater BC production cost in contrast VC’s is another disadvantage, but that can be reduced with the use of low-cost waste products and byproducts as substrates, such as sugarcane molasses, distillery effluents, crude glycerol derived from biofuel, lignocellulosic residues, etc. The use of mixed raw materials rather than a solemnly synthetic reagents has the potential to augment the productivity of BC due to synergic effects and diminish the likelihood of the unavailability of substrate. Conventional carbon sources used during the bacterial fermentation include dextrose, monosaccharides, and polyol compounds, which remarkably increase the expenditure, accounting for approximately 30% of the total cost, and has the potential of even better results. Greater effort should be directed at the functionalization of BC to expand its applications in diverse industries.

Another factor that should be considered is the biosafety of the material, especially when used as a wound dressing, since the material is in direct contact with wounds and even deeper layers of the dermis. Therefore, the purification and sterilization of BC membranes are of utmost importance.

Other obstacles must also be the overcome for the broad use of BC in the health field. However, the current abundance of studies and patents and the use of safe sterilization techniques, such as irradiation for the sterilization of membranes, indicates a near implementation of this interesting material in versatile and advanced biotechnological applications throughout the world. 

## 7. Conclusions

Disposable synthetic products used in medicine are responsible for a great deal of waste worldwide, including air and soil pollution. Therefore, research has focused on sustainable provisions as an alternative to the conventionally used materials. Plenty of materials have been gaining attention, mainly natural polymers, such as chitin, cellulose, starch, and others, because of their versatility. A controlled modification of such materials was performed with the aim of a safe use in medicine. BC is a naturally produced polymer with a great ease of controllability. This review article focused on BC production and its interesting applications in the biomedical field. In contrast to VC and synthetic polymers, the microbial polymer appears as a great asset because of its singular and versatile properties, including its great mechanical strength, high WHC, high WRR, high crystallinity, permeation of gas, high purity, and others.

The structural morphology of BC is the main reason for its use as a material in the medical field, which enables the stability of wound dressings. BC also offers biocompatibility, non-toxicity, the protection of wounds from bacterial invasion, the assurance of thermal and gas exchange, the provision of an ideal environment for the skin’s acceleration recovery process, and the adsorption of excess exudate from the wound. BC also constitutes a more economical option compared to conventional dressings. The numerous studies and patents demonstrate BC’s potential for a broad gamut of applications, which includes drug delivery, tissue engineering, and other uses in the pharmaceutical and medical fields.

## Figures and Tables

**Figure 1 molecules-27-05580-f001:**
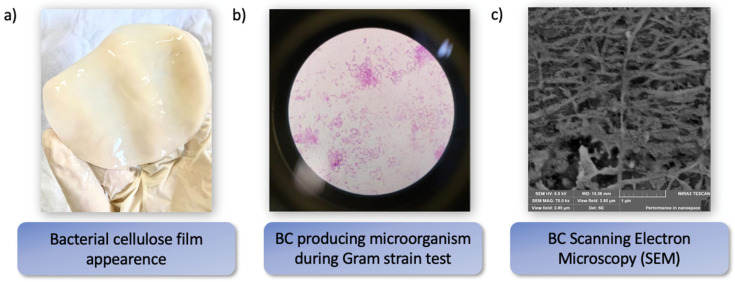
(**a**) Image of a bacterial cellulose film; (**b**) Microscopic image of *Komagataeibacter hansenii* during Gram strain test; (**c**) Scanning Electron Microscopy (SEM) of *Komagataeibacter hansenii* nanofibrils.

**Figure 2 molecules-27-05580-f002:**
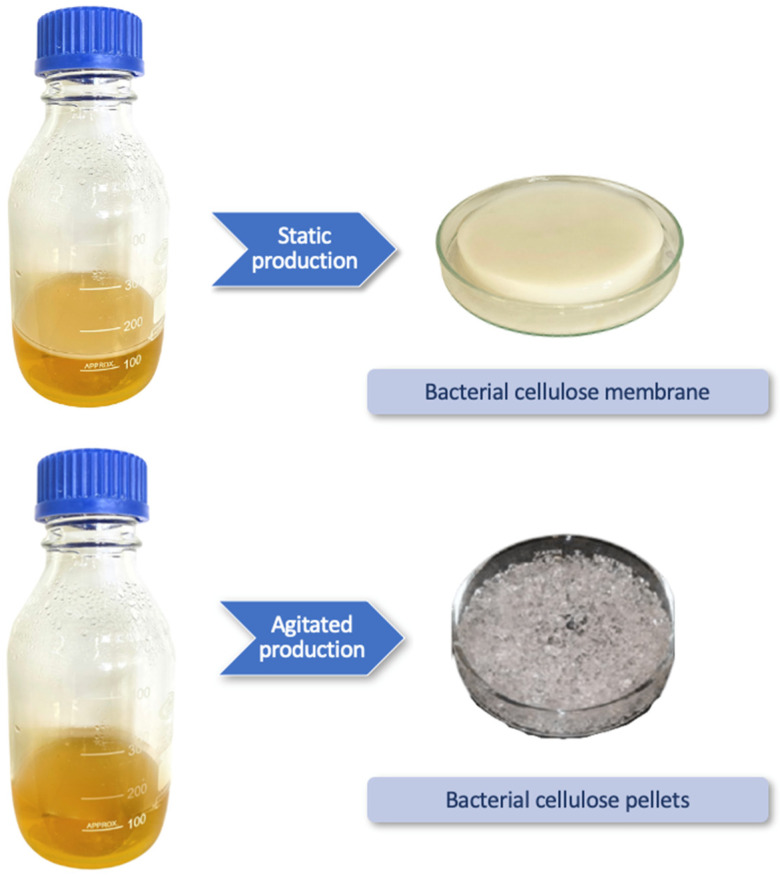
Bacterial cellulose appearance in static and agitated manners.

**Figure 3 molecules-27-05580-f003:**
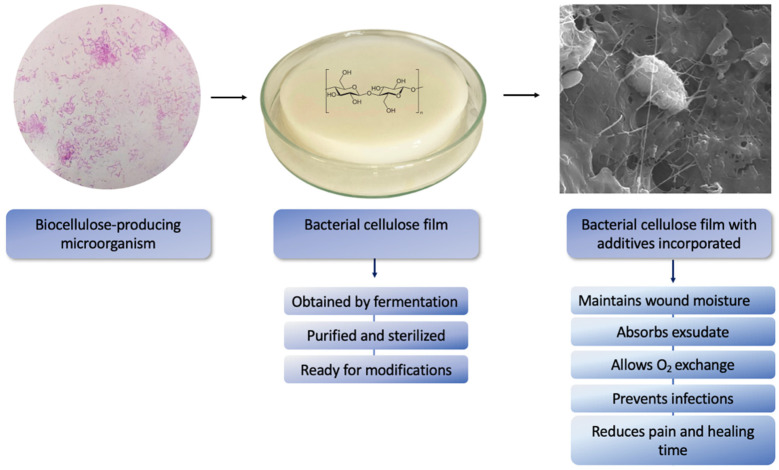
Schematic synthesis of BC film by cellulose-producing microorganism followed by production of bacterial cellulose with additive incorporated and characteristics as wound dressing.

**Figure 4 molecules-27-05580-f004:**
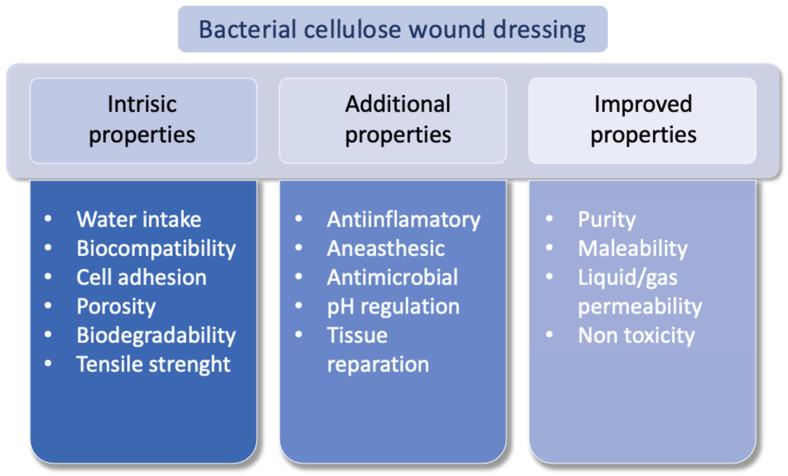
Additional, intrinsic, and improved properties of bacterial cellulose for wound dressing. Modified from Khan et al. [68].

**Figure 5 molecules-27-05580-f005:**
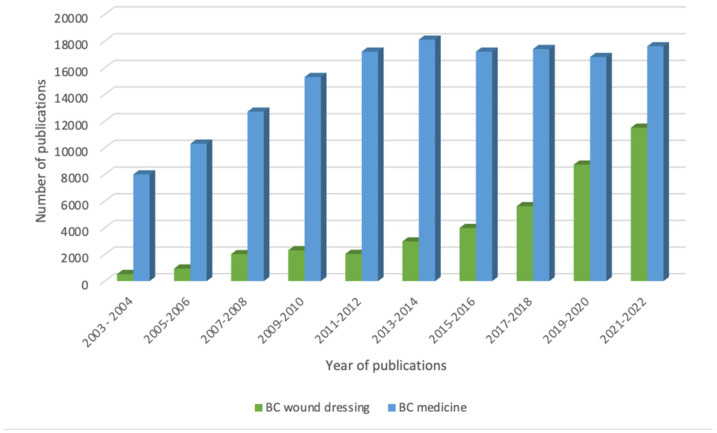
Bacterial cellulose publications (Google Scholar search system, with the descriptors: ‘bacterial cellulose’ and ‘medicine’ and ‘bacterial cellulose’ and ‘wound dressing).

**Table 1 molecules-27-05580-t001:** Commonly used production methods for bacterial cellulose.

Production Method Production	Fermentation Characteristics	Bacterial Cellulose Appearence
Static production	Predominantly used on laboratory scale; Fermentation process up to two weeks [86].	Homogenous film/membrane
Agitated production	Increase in O_2_ delivery to microorganism; May result in lower production yield [87].	Pellets
Airlift bioreactor (ABR) production	Increase in O_2_ delivery to microorganism [11].	Pellets
Rotating disc bioreactor (RDB) production	Yield similar to that of static production [88].	Homogenous film/membrane
Trickle bed reactor (TBR) production	Increase in O_2_ delivery to microorganism; Lower sheer force [89].	Irregular cellulose membrane

**Table 3 molecules-27-05580-t003:** Bacterial cellulose commercial byproducts available on the market for wound dressing applications.

Product	Application	Company/Usage
Dermafill™	Burns	Robin Goad, USA
Bionext^®^	Burns, ulcers and lacerations	Bionext Produtos Biotecnologicos, Brazil
Prima Cel™	Ulcers	Xylos Corporation, USA
Bioprocess^®^	Burns	Bio Fill Produtos Biotecnologicos, Brazil
Xcell^®^	Venous ulcer wounds	XCELL BIOLOGIX, USA
MTA protective tissue	Injury and wound care	Xylos Corporation, USA

## Data Availability

Not applicable.

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
