# Peer review of "Bacterial Cellulose as a Versatile Biomaterial for Wound Dressing Application"

_molecules, 2022, doi:10.3390/molecules27175580_

Round 1
Reviewer 1 Report
In this review, authors have discussed the use of bacterial cellulose in development of wound dressing materials for treating skin-related issues caused by injuries and microbial infections. They attempted to summarize the methods used for BC production, developing BC-based composites with a variety of materials, and finally discussing the applications of BC-based composites in biomedical field. In present form, this review is not suitable for publication and needs extensive revision, according to the below comments and suggestions, for reconsideration.
[1] There are many reviews on using Bacterial Cellulose for wound dressing applications. It is suggested to provide a comparison with those reviews, probably in Introduction, and emphasize how this review is different from the published ones.
- Microbial Biotechnology, 2019;12(4):586-610.
- Current Pharmaceutical Design, 2022;28(7):570-580.
- Biotechnology Advances, 2015;33(8):1547-71
- Polymers (Basel). 2021 Feb; 13(3): 412
- Frontiers in Bioengineering and Biotechnology, 2020, 23;8:593768
[2] Title: You used the term ‘biocellulose’ which is misleading as you only focused on ‘bacterial cellulose’ and not all kinds of celluloses obtained from natural sources like algae, animals, and plant sources. Therefore, it is suggested to replace it with ‘bacterial cellulose’. Furthermore, you only mentioned the use of cellulose as ‘wound dressing’ but then you have added separate section on ‘drug delivery’. Although it is understandable that both applications, i.e., wound healing and drug delivery’ go in parallel in most of the cases, the title is little misleading, and should be revised according to the contents covered in this review.
[3] Abstract and Introduction: You mentioned that this review describes different methods of BC production and its modification, but apparently there are no separate sections about these aspects, rather these are just described superficially. It is suggested to add separate sections on production methods and modification approaches.
[4] It is strongly suggested to add a section that summarizes the salient features of BC making it a suitable material for wound dressing applications.
[5] Section 2 ‘Wound dressings’: It is suggested to discuss the wound healing mechanism. For instance, describe the four stages (hemostasis, inflammation, proliferation, and remodeling) and events (Coagulation, Inflammation, Migration, Division of parenchymal cells and connective tissues, Angiogenesis, Development of extracellular matrix) involved in the healing process (https://doi.org/10.3389/fbioe.2020.553037).
[6] Table 2. On what basis you selected the provided studies? I would rather suggest to reorganize the table and cover the contents like BC, modification method, improved properties, application, and references.
[7] Line 46: what do you mean by ‘free of environmental dependence’?
[8] A major issue with this review is that the authors have only summarized the studies on conventional approaches of developing BC-based wound dressings. It is suggested to discuss the current trends of developing BC-based wound dressings, like BC-based transparent wound dressings (https://doi.org/10.1016/j.jddst.2017.12.019; https://doi.org/10.1016/j.ijbiomac.2017.07.075), BC-based electroactive wound dressings (https://doi.org/10.1016/j.cej.2021.130563), those with anti-inflammatory capabilities (https://doi.org/10.1002/adhm.202100402), development of green composites for biomedical applications (Advanced Composites and Hybrid Materials, 2022, 5:307-321).
[9] Headings 5.2.2 and 5.2.3 are same. How the two sections are different from each other?
[10] Lines 64-65: this sentence is identical to the opening sentence of the Abstract, and should be rephrased.
[11] Lines 94-95: It is suggested to refine the definition of biocompatibility. ‘Biocompatibility refers to the ability of a material to interact with living tissues without provoking the immunogenic response, inflammation, or allergy and remain non-toxic’ (Frontiers in Bioengineering and Biotechnology. 9,2021:616555). Furthermore, the focus in this section should be describing the biocompatibility of BC, both alone and in the form of composites with other biocompatible materials. There is only one sub-section under Section, which is inappropriate.
[12] Section 5 ‘Bacterial cellulose’: No need to talk about plant cellulose again here. Some additional suggestions to this section are:
- Line 207: Provide a reference for production of cellulose by the ‘cell-free enzyme systems’. This is commendable as this is a recent approach developed by Park and co-workers for cellulose production. Additionally, it is further suggested to add a little explanation to cell-free synthesis of cellulose like self-assembly of cellulose nanofibrils through intermediate phases and how structural and physico-mechanical and other properties of cellulose produced by the cell-free enzyme system are different from that produced by the microbial cell system.
- Lines 219-225: A recent publication provided insights into the molecular regulation of the extracellular transport of cellulose fibrils, their organization into high order supramolecular structures, and functionalization for multipurpose applications (Progress in Materials Science, 129: 100972, 2022).
[13] Table 1. This table should be reorganized by describing the appearance, properties, yield, incubation time and conditions, etc. for each method of BC production.
[14] Lines 527-529: Here the authors mentioned about the high production cost of BC as a disadvantage, that is true. I would suggest to add a brief discussion on the economic analysis of BC production in terms of using low cost substrate. Please refer to the manuscript of Ul-Islam et al., for such analysis (https://doi.org/10.1007/s11814-020-0524-3)
[15] Figure 1, 2, and 3 are very basic figures and don’t provide any unique information. It is suggested to replace these figures with some appealing ones.
[16] Figure 8. It appears that the presented figure is modified from a figure in ref. [41] that actually classified the properties of BC into ‘intrinsic properties’, ‘improved properties’, and ‘added properties’ with slight modifications. The original source should be acknowledged. Same for figure 6 (Biotechnology Advances, 2015;33(8):1547-71).
[17] Figure 4 might not be very important as it summarizes applications of BC in different fields while this review is mainly focused on the wound dressing applications of BC.
[18] Lines 264-265: provide specific reference for each mentioned chemical modification method.
[19] Section 5.1, lines 256-258: BC is also used in development of advanced biosensing materials (https://doi.org/10.1016/j.ijbiomac.2020.10.217; https://doi.org/10.1016/j.bios.2020.112163) and diagnosis (Development of three-dimensional bacterial cellulose/chitosan scaffolds: Analysis of cell-scaffold interaction for potential application in the diagnosis of ovarian cancer. International Journal of Biological Macromolecules, 137:1050–1059, 2019).
[20] There are many small passages, with only 2-3 lines. All small passages should be merged together and there should be a flow of information.
[21] Line 287: What do you mean by ‘derived compounds’ of BC?
[22] Line 275: Pristine BC does not support the adhesion of cells due to lack of adhesive sites. A correction is needed.
[23] In general, manuscript lacks discussion of the cited literature at many places. Authors need to provide a rational discussion of each cited literature, where possible.
[24] Take care of italicization: in vitro, in vivo, in situ, ex situ, via, etc.
[25] Abbreviations should be defined at first appearance and then used consistently. For example, bacterial cellulose is defined at line 45, but then full form is present at many occasions.
[26] There are several grammatical errors and typos throughout the manuscript. Therefore, a careful proofread is required at the revision stage.
Author Response
Dear Reviewer,
We are very grateful for the positive and helpful suggestions and expert comments. We feel that the quality of the manuscript has been improved as a result. We would like to take this opportunity to express our sincere thanks to you who identified areas of our manuscript that needed corrections or modification. We next detail our responses to yours concerns and comments. Modifications in the manuscript are in red. We hope that these revisions improve the paper such that you now deem it worthy of publication in Molecules.
[1] There are many reviews on using Bacterial Cellulose for wound dressing applications. It is suggested to provide a comparison with those reviews, probably in Introduction, and emphasize how this review is different from the published ones.
- Microbial Biotechnology, 2019;12(4):586-610.
- Current Pharmaceutical Design, 2022;28(7):570-580.
- Biotechnology Advances, 2015;33(8):1547-71
- Polymers (Basel). 2021 Feb; 13(3): 412
- Frontiers in Bioengineering and Biotechnology, 2020, 23;8:593768
RESPONSE Those are very interesting reviews. Our review focused on the use of novelty techniques that were not cited by these papers, as in the present year, the number of publications has grown significantly, and with that, there is a need for new reviews synthesizing the innovative research on the subject. However, as also requested by the reviewer in another comment, additional innovative trends and techniques were also covered and these references were also included in the manuscript.
[2] Title: You used the term ‘biocellulose’ which is misleading as you only focused on ‘bacterial cellulose’ and not all kinds of celluloses obtained from natural sources like algae, animals, and plant sources. Therefore, it is suggested to replace it with ‘bacterial cellulose’. Furthermore, you only mentioned the use of cellulose as ‘wound dressing’ but then you have added separate section on ‘drug delivery’. Although it is understandable that both applications, i.e., wound healing and drug delivery’ go in parallel in most of the cases, the title is little misleading, and should be revised according to the contents covered in this review.
RESPONSE Thank you very much for the remark. The use of the term 'biocellulose' has been corrected. The title has also been changed, according to your suggestion, to 'Bacterial Cellulose’s Versatility in Medicine: A Review on Wound Dressing Materials.'
[3] Abstract and Introduction: You mentioned that this review describes different methods of BC production and its modification, but apparently there are no separate sections about these aspects, rather these are just described superficially. It is suggested to add separate sections on production methods and modification approaches.
RESPONSE ​​A section '5.1. Bacterial cellulose synthesis and production' has been added to the review, and more information has been provided. Thank you for the suggestion.
[4] It is strongly suggested to add a section that summarizes the salient features of BC making it a suitable material for wound dressing applications.
RESPONSE We really appreciate the suggestion. New information has been added and we believe that the salient features of BC have been summarized throughout the manuscript.
[5] Section 2 ‘Wound dressings’: It is suggested to discuss the wound healing mechanism. For instance, describe the four stages (hemostasis, inflammation, proliferation, and remodeling) and events (Coagulation, Inflammation, Migration, Division of parenchymal cells and connective tissues, Angiogenesis, Development of extracellular matrix) involved in the healing process (https://doi.org/10.3389/fbioe.2020.553037).
RESPONSE That is a very pertinent suggestion! Thank you, the wound healing mechanism was added to the review.
[6] Table 2. On what basis you selected the provided studies? I would rather suggest to reorganize the table and cover the contents like BC, modification method, improved properties, application, and references.
RESPONSE The provided studies were selected according to the relevance of the paper on the subject according to Science Direct. Table 2 has been reorganized. And a graph (Figure 8) has been added with more detailed information regarding the number of publications on the area.
[7] Line 46: what do you mean by ‘free of environmental dependence’?
RESPONSE The term has been wrongly used. It has been removed.
[8] A major issue with this review is that the authors have only summarized the studies on conventional approaches of developing BC-based wound dressings. It is suggested to discuss the current trends of developing BC-based wound dressings, like BC-based transparent wound dressings (https://doi.org/10.1016/j.jddst.2017.12.019; https://doi.org/10.1016/j.ijbiomac.2017.07.075), BC-based electroactive wound dressings (https://doi.org/10.1016/j.cej.2021.130563), those with anti-inflammatory capabilities (https://doi.org/10.1002/adhm.202100402), development of green composites for biomedical applications (Advanced Composites and Hybrid Materials, 2022, 5:307-321).
RESPONSE The suggested studies have been added to Table 2.
[9] Headings 5.2.2 and 5.2.3 are same. How the two sections are different from each other?
RESPONSE This was a typing error, we are sorry. Heading 5.2.2. Is supposed to be 'Bacterial Cellulose in Medicine', this has been corrected.
[10] Lines 64-65: this sentence is identical to the opening sentence of the Abstract, and should be rephrased.
RESPONSE Thank you for the remark! The sentence has been rewritten.
[11] Lines 94-95: It is suggested to refine the definition of biocompatibility. ‘Biocompatibility refers to the ability of a material to interact with living tissues without provoking the immunogenic response, inflammation, or allergy and remain non-toxic’ (Frontiers in Bioengineering and Biotechnology. 9,2021:616555). Furthermore, the focus in this section should be describing the biocompatibility of BC, both alone and in the form of composites with other biocompatible materials. There is only one sub-section under Section, which is inappropriate.
RESPONSE Further information regarding BC's biocompatibility was added to the new subsections (3.1. Bacterial cellulose biocompatibility and 3.2. Bacterial cellulose composites biocompatibility).
[12] Section 5 ‘Bacterial cellulose’: No need to talk about plant cellulose again here. Some additional suggestions to this section are:
- Line 207: Provide a reference for production of cellulose by the ‘cell-free enzyme systems’. This is commendable as this is a recent approach developed by Park and co-workers for cellulose production. Additionally, it is further suggested to add a little explanation to cell-free synthesis of cellulose like self-assembly of cellulose nanofibrils through intermediate phases and how structural and physico-mechanical and other properties of cellulose produced by the cell-free enzyme system are different from that produced by the microbial cell system.
- Lines 219-225: A recent publication provided insights into the molecular regulation of the extracellular transport of cellulose fibrils, their organization into high order supramolecular structures, and functionalization for multipurpose applications (Progress in Materials Science, 129: 100972, 2022).
RESPONSE The plant cellulose has been removed from Section 5. The requested references and a brief explanation on the differences of cell-free enzyme systems and microbial cell systems have been provided.
[13] Table 1. This table should be reorganized by describing the appearance, properties, yield, incubation time and conditions, etc. for each method of BC production.
RESPONSE A new column with ' Bacterial cellulose appearance' has been added. As for the other columns suggestions, we believe that as the works use different production media, conditions and fermentation times, this could make the time a little confusing.
[14] Lines 527-529: Here the authors mentioned about the high production cost of BC as a disadvantage, that is true. I would suggest to add a brief discussion on the economic analysis of BC production in terms of using low cost substrate. Please refer to the manuscript of Ul-Islam et al., for such analysis (https://doi.org/10.1007/s11814-020-0524-3).
RESPONSE The reference and discussion have been added, as suggested.
[15] Figure 1, 2, and 3 are very basic figures and don’t provide any unique information. It is suggested to replace these figures with some appealing ones.
RESPONSE Thank you for the suggestion. Figure 3 has been replaced.
[16] Figure 8. It appears that the presented figure is modified from a figure in ref. [41] that actually classified the properties of BC into ‘intrinsic properties’, ‘improved properties’, and ‘added properties’ with slight modifications. The original source should be acknowledged. Same for figure 6 (Biotechnology Advances, 2015;33(8):1547-71).
RESPONSE Thank you for the observation, we are sorry about this confusion. The inspirational reference has been added for Figure 8. Figure 6 has been removed and summarized through the text.
[17] Figure 4 might not be very important as it summarizes applications of BC in different fields while this review is mainly focused on the wound dressing applications of BC.
RESPONSE Figure 3 has been removed.
[18] Lines 264-265: provide specific reference for each mentioned chemical modification method.
RESPONSE The reference has been added.
[19] Section 5.1, lines 256-258: BC is also used in development of advanced biosensing materials (https://doi.org/10.1016/j.ijbiomac.2020.10.217; https://doi.org/10.1016/j.bios.2020.112163) and diagnosis (Development of three-dimensional bacterial cellulose/chitosan scaffolds: Analysis of cell-scaffold interaction for potential application in the diagnosis of ovarian canceResponse International Journal of Biological Macromolecules, 137:1050–1059, 2019).
RESPONSE The references have been added.
[20] There are many small passages, with only 2-3 lines. All small passages should be merged together and there should be a flow of information.
RESPONSE Thank you for the remark, this has been carefully corrected.
[21] Line 287: What do you mean by ‘derived compounds’ of BC?
RESPONSE It was meant as BC's products, the sentence was rewritten for a better understanding.
[22] Line 275: Pristine BC does not support the adhesion of cells due to lack of adhesive sites. A correction is needed.
RESPONSE This information has been corrected. We appreciate the observation.
[23] In general, manuscript lacks discussion of the cited literature at many places. Authors need to provide a rational discussion of each cited literature, where possible.
RESPONSE Thank you for your feedback, we believe that the manuscript is now more robust.
[24] Take care of italicization: in vitro, in vivo, in situ, ex situ, via, etc.
RESPONSE This has been taken care of.
[25] Abbreviations should be defined at first appearance and then used consistently. For example, bacterial cellulose is defined at line 45, but then full form is present at many occasions.
RESPONSE We are sorry for this mistake. This was corrected.
[26] There are several grammatical errors and typos throughout the manuscript. Therefore, a careful proofread is required at the revision stage.
RESPONSE The manuscript has been carefully revised, as suggested. Many thanks.
Reviewer 2 Report
<Biocellulose Applied as Wound Dressing: A Review> is an interesting paper in which are presented the advances in the BC use in the medical field, especially as wound healing, engineering scaffolds and drug delivery systems. The advantages and disadvantages of using BC are highlighted, the emphasis being focused on the possibilities to remove the disadvantages of BC. The figures are suggestive and reinforce the written text. The text is cursive and clear and it is supported by many references. In conclusion, the paper could be published after minor revision.
I have found only minor errors:
Page 7/line 245/are excellent features for biomedical applications.40
Page 8/line 285/the incorporation of propolis extract5
Page 14/Line 461/ 5.2.2. Bacterial Cellulose for drug delivery
Page 14/line 492/ 5.2.3. Bacterial Cellulose for drug delivery
The paragraph 5.2.3 title must be change.
Author Response
Page 7/line 245/are excellent features for biomedical applications.40
RESPONSE Thank you, this has been corrected.
Page 8/line 285/the incorporation of propolis extract5
RESPONSE Thank you, this has been corrected.
Page 14/Line 461/ 5.2.2. Bacterial Cellulose for drug delivery
Page 14/line 492/ 5.2.3. Bacterial Cellulose for drug delivery
The paragraph 5.2.3 title must be change.
RESPONSE This was a typing error, we are sorry. Heading 5.2.2. Is supposed to be 'Bacterial Cellulose in Medicine', this has been corrected.
Reviewer 3 Report
The manuscript "Biocellulose Applied as Wound Dressing: A Review" aims to describe the state of the art of the bacterial cellulose applications, with focus on wound dressing capabilities. It is a valuable study that can be published after authors address the following problems:
The title should be changed to “Bacterial cellulose applied as ....” (all the cellulose is bio!).
The introduction section must be reworked. Why authors chosen bacterial cellulose as subject? Authors must explain better the motivation behind chosen this system. A logical setup would be that after the current starting point (cellulose – limits - bacterial cellulose – advantages) to continue with:
- specific uses (not only wound dressing, but also other antimicrobial or medical uses for example food packaging, membranes, scaffolds etc – doi: 10.3390/polym13223950; doi: 10.1080/1539445X.2021.1944208; doi: 10.3390/pharmaceutics13050613) – followed by,
- chosen application – wound dressing – comparison with other systems, especially those based also on polysaccharides like chitosan, alginate, starch, pullulan etc (doi: 10.3390/pharmaceutics12100983; doi: 10.3390/polym14040799; doi: 10.3390/nano11092377);
- more recent literature on bacterial cellulose and its applications then follows. The authors need to update the introduction by citing following doi: 10.3390/molecules25184069; doi: 10.3390/ma13214793; doi: 10.3390/ma15031054.
While the subject of this review is clear (BC – wound dressing), authors should indicate also the review methodology (what databases they searched, what keywords, what time interval etc.) Is imperative to know the limits of the review.
The English language needs some polishing for style and typos (e.g. an extra 40 at row 245, a 5 at row 285)
Multiple references like 94, 95, 96 should be condensed 94-96 (row 289).
Please use the proper notation for measurement units: kwh/ton is with capital W as it abbreviates a person name.
Authors should specify (around rows 100-110, or develop the idea from rows 121-122, 125-127) that biomaterials are biocompatible as a function of specific applications. Biocompatibility is not a given property for a biomaterial (e.g. a material that is biocompatible in bone grafting might not be biocompatible in cardiovascular surgery).
Author Response
Dear Reviewer,
We are very grateful for the positive and helpful suggestions and expert comments. We feel that the quality of the manuscript has been improved as a result. We would like to take this opportunity to express our sincere thanks to you who identified areas of our manuscript that needed corrections or modification. We next detail our responses to yours concerns and comments. Modifications in the manuscript are in red. We hope that these revisions improve the paper such that you now deem it worthy of publication in Molecules.
The manuscript "Biocellulose Applied as Wound Dressing: A Review" aims to describe the state of the art of the bacterial cellulose applications, with focus on wound dressing capabilities. It is a valuable study that can be published after authors address the following problems:
The title should be changed to “Bacterial cellulose applied as ....” (all the cellulose is bio!).
RESPONSE Thank you very much for your feedback. The title has been changed, according to your suggestion, to 'Bacterial Cellulose’s Versatility in Medicine: A Review on Wound Dressing Materials.'
The introduction section must be reworked. Why authors chosen bacterial cellulose as subject? Authors must explain better the motivation behind chosen this system. A logical setup would be that after the current starting point (cellulose – limits - bacterial cellulose – advantages) to continue with:
- specific uses (not only wound dressing, but also other antimicrobial or medical uses for example food packaging, membranes, scaffolds etc – doi: 10.3390/polym13223950; doi: 10.1080/1539445X.2021.1944208; doi: 10.3390/pharmaceutics13050613) – followed by,
- chosen application – wound dressing – comparison with other systems, especially those based also on polysaccharides like chitosan, alginate, starch, pullulan etc (doi:10.3390/pharmaceutics12100983; doi: 10.3390/polym14040799; doi: 10.3390/nano11092377);
- more recent literature on bacterial cellulose and its applications then follows. The authors need to update the introduction by citing following doi: 10.3390/molecules25184069; doi: 10.3390/ma13214793; doi: 10.3390/ma15031054.
RESPONSE The introduction has been reworked, as requested. A brief comparison with other polysaccharides has also been added in the '4.1. Biopolymers in medicine' subheading. And all the suggested papers have been added to the manuscript. We hope that such changes have fulfilled your observations.
While the subject of this review is clear (BC – wound dressing), authors should indicate also the review methodology (what databases they searched, what keywords, what time interval etc.) Is imperative to know the limits of the review.
RESPONSE A review methodology has been added in the introduction of the manuscript and a graph (Figure 6) has been added with more detailed information regarding the number of publications in the area. We really appreciate the suggestion!
The English language needs some polishing for style and typos (e.g. an extra 40 at row 245, a 5 at row 285)
RESPONSE We've done a careful revision, and this has been corrected.
Multiple references like 94, 95, 96 should be condensed 94-96 (row 289).
RESPONSE Thank you for the observation! This has been corrected.
Please use the proper notation for measurement units: kwh/ton is with capital W as it abbreviates a person name.
RESPONSE The unit has been rewritten.
Authors should specify (around rows 100-110, or develop the idea from rows 121-122, 125-127) that biomaterials are biocompatible as a function of specific applications. Biocompatibility is not a given property for a biomaterial (e.g. a material that is biocompatible in bone grafting might not be biocompatible in cardiovascular surgery).
RESPONSE This has been summarized in the last paragraph of the heading '3. Biocompatibility', with the given example of ​​poly(lactic-co-glycolic acid). However, this heading has been rewritten for a better understanding.
Round 2
Reviewer 1 Report
Although authors have substantially improved the text of the manuscript, there are still a few issues mainly in the structure of manuscript. Therefore, the manuscript needs another round of revision according to below comments, before publications.
1. The title still needs refinement. A title is suggested as below
‘Bacterial cellulose as a versatile biomaterial for wound dressing application’
2. The different sections in the manuscript should be reorganized as follow:
- Introduction
- Wound dressing
- Biopolymers and their use in medicine
- Biocompatibility of polymers for medical applications
- BC synthesis and structural properties
- Applications of BC in wound dressing
- Biocompatibility of BC for wound healing
- Conclusion and future perspective
3. Authors did not improve the figure as suggested before. None of the provided figure is appealing for the readers.
4. Combine Figure 1, 2, and 3, also cite the source of SEM image in the figure legend.
5. Move Figure 5 to section ‘Biopolymers and their use in medicine’.
6. As suggested previously, cite the source of Figure 7 in figure legend.
7. List of articles provided in Table 2 is unnecessary, just leave the list of patents and remove the list of studies.
8. For figure 8, authors should collect the publication data from Clarivate (Web of Science) since Sciencedirect has limited collection of publications on the topic.
Author Response
- The title still needs refinement. A title is suggested as below ‘Bacterial cellulose as a versatile biomaterial for wound dressing application’.
Response: Thank you very much. The title has been changed according to the suggestion.
- The different sections in the manuscript should be reorganized as follow:
- Introduction
- Wound dressing
- Biopolymers and their use in medicine
- Biocompatibility of polymers for medical applications
- BC synthesis and structural properties
- Applications of BC in wound dressing
- Biocompatibility of BC for wound healing
- Conclusion and future perspective
Response: We truly appreciate this suggestion, and we think that this is also a good organization for this manuscript. However, all authors believe that the sections should remain with the same organization, as it is our intent to first reference general medical information and then to focus on more specific subjects proposed by the article such as BC in wound dressing and drug delivery systems. ​​We hope the reviewer can accept our decision. Many thanks.
- Authors did not improve the figure as suggested before. None of the provided figure is appealing for the readers.
Response: The figures have been improved again. We hope that this made them more appealing.
- Combine Figure 1, 2, and 3, also cite the source of SEM image in the figure legend.
Response: Figures 1, 2 and 3 are all unpublished images from the authors. They have been combined and been replaced.
- Move Figure 5 to section ‘Biopolymers and their use in medicine’.
Response: The figure has been moved.
- As suggested previously, cite the source of Figure 7 in figure legend.
Response: The source has been cited in the legend.
- List of articles provided in Table 2 is unnecessary, just leave the list of patents and remove the list of studies.
Response: The list or articles have been removed, as suggested and they have been cited throughout the same subheading, as the authors believe that those are important works that still needs to be cited.
- For figure 8, authors should collect the publication data from Clarivate (Web of Science) since Sciencedirect has limited collection of publications on the topic.
Response. Thank you very much for the suggestion, the figure has been updated according to Clarivate data.
Reviewer 3 Report
The authors have responded to my comments and have addressed all my concerns, substantially improving the manuscript, therefore, I suggest publishing the paper in the current form.
Author Response
Dear Reviewer,
Many thanks.